# Current Status and Future Prospects for Shared Decision Making before and after Total Knee Replacement Surgery—A Scoping Review

**DOI:** 10.3390/ijerph18020668

**Published:** 2021-01-14

**Authors:** Geert van der Sluis, Jelmer Jager, Ilona Punt, Alexandra Goldbohm, Marjan J. Meinders, Richard Bimmel, Nico L.U. van Meeteren, Maria W. G. Nijhuis-van Der Sanden, Thomas J. Hoogeboom

**Affiliations:** 1Department of Health Strategy and Innovation, Nij Smellinghe Hospital Drachten, Compagnonsplein 1, 9202 NN Drachten, The Netherlands; 2Department of Epidemiology, Care and Public Health Research Institute (CAPHRI), School for Public Health and Primary Care, Maastricht University, Universiteitssingel 40, 6229 ER Maastricht, The Netherlands; jelmer.jager@olvg.nl; 3Department of Physical Therapy, Onze Lieve Vrouwe Gasthuis (OLVG), Hospital Amsterdam, Oosterpark 9, 1091 AC Amsterdam, The Netherlands; 4Faculty of Health, University of Applied Sciences Leiden, Zernikedreef 11, 2333 CK Leiden, The Netherlands; 5Department of Orthopaedics, Maastricht University Medical Centre, P. Debyelaan 25, 6229 HX Maastricht, The Netherlands; ilona.punt@maastrichtuniversity.nl; 6Department of Surgery and Trauma Surgery and Research School NUTRIM, Maastricht University and Maastricht University Medical Centre, 6229 HX Maastricht, The Netherlands; 7Retired, 0320 Lelystad, The Netherlands; vandersluis.geert@gmail.com; 8Radboud University Medical Center, Radboud Institute for Health Sciences, IQ Healthcare, Geert Grooteplein 21, 6525 EZ Nijmegen, The Netherlands; Marjan.Meinders@radboudumc.nl (M.J.M.); ria.nijhuis-vandersanden@radboudumc.nl (M.W.G.N.-v.D.S.); Thomas.Hoogeboom@radboudumc.nl (T.J.H.); 9Department of Orthopedics and Traumatology, Nij Smellinghe Hospital Drachten, Compagnonsplein 1, 9202 NN Drachten, The Netherlands; r.bimmel@nijsmellinghe.nl; 10Topsector Life Sciences and Health (Health~Holland), Laan van Nieuw Oost-Indie 334, 2693 CE the Hague, The Netherlands; meeteren@health-holland.com; 11Department of Anesthesiology, Erasmus Medical Center, Doctor Molewaterplein 40, 3015 GD Rotterdam, The Netherlands

**Keywords:** shared decision making, total knee replacement, patient-centered care

## Abstract

*Background*. To gain insight into the current state-of-the-art of shared decision making (SDM) during decisions related to pre and postoperative care process regarding primary total knee replacement (TKR). *Methods*. A scoping review was performed to synthesize existing scientific research regarding (1) decisional needs and preferences of patients preparing for, undergoing and recovering from TKR surgery, (2) the relation between TKR decision-support interventions and SDM elements (i.e., team talk, option talk, and decision talk), (3) the extent to which TKR decision-support interventions address patients’ decisional needs and preferences. *Results*. 2526 articles were identified, of which 17 articles met the inclusion criteria. Of the 17 articles, ten had a qualitative study design and seven had a quantitative study design. All included articles focused on the decision whether to undergo TKR surgery or not. Ten articles (all qualitative) examined patients’ decisional needs and preferences. From these, we identified four domains that affected the patients’ decision to undergo TKR: (1) personal factors, (2) external factors, (3) information sources and (4) preferences towards outcome prediction. Seven studies (5) randomized controlled trials and 2 cohort studies) used quantitative analyses to probe the effect of decision aids on SDM and/or clinical outcomes. In general, existing decision aids did not appear to be tailored to patient needs and preferences, nor were the principles of SDM well-articulated in the design of decision aids. *Conclusions.* SDM in TKR care is understudied; existing research appears to be narrow in scope with limited relevance to established SDM principles and the decisional needs of patients undertaking TKR surgery.

## 1. Introduction

Across healthcare, shared decision making (SDM) is increasingly considered as the preferable method of arriving at clinical decisions [1]. Different theoretical models of SDM exist. One of the well-established models was described by Elwyn et al. (2017) [2]. Elwyn and his co-authors defined SDM as a process in which decisions are made in a collaborative way, where trustworthy information about a set of options is provided to patients in an accessible format, typically in situations where the preferences, values and individual circumstances of patients and their families play a major role in decisions [2,3]. The application of SDM in clinical practice as proposed by Elwyn et al. should involve three major elements: (1) Team talk, (2) Option talk and (3) Decision talk [2]. Team talk puts emphasis on supporting patients as they are made aware of choices, while also eliciting their goals as a means of guiding the decision-making process. *Option talk* refers to the task of comparing treatment options while highlighting the relative pros and cons of each option. *Decision talk* refers to the process of allowing for deliberation over specific healthcare options while explicitly respecting the preferences of patients [2]. SDM is considered most valuable if more than one reasonable path forward exists [1]. 

A clear example of a clinical situation in which SDM is of potential value is the decision to undertake elective total knee replacement (TKR) surgery. Despite the fact that this procedure is very common, long-term outcomes are not always optimal for everyone. For example, Hawker et al., (2013) demonstrated that half of frail older adults did not experience a clinically meaningful improvement in function following TKR [4], while Beswick et al., (2012) reported that up to 34% of people with TKR experienced moderate to severe chronic pain, even after full recovery should have been achieved [5]. As there are a number of reasonable alternatives to TKR (e.g., exercise interventions [6,7], joint distraction [8], one possible route to better clinical outcomes is to improve the decision-making process prior to surgery, to ensure that candidates have realistic expectations about the outcomes. Additionally, myriad decisions are encountered in the preparation for surgery, as well as during postoperative recovery and rehabilitation. According to SDM principles, the healthcare professionals and patients should ideally weigh available treatment options together while taking patient values and preferences into consideration related to the entire TKR healthcare process from preparation to surgery to rehabilitation. 

To date, little is known about the state of SDM research in TKR surgery and particularly the extent to which existing research relates to underlying SDM elements or the current best understanding of patients’ decisional needs and preferences. Therefore, the purpose of this scoping review is to probe the available literature in order to synthesize what is known regarding both patient thoughts and preferences surrounding TKR as well as the state of the art of SDM in TKR care. We formulated the following specific research questions (RQ’s):
What is known regarding the decisional needs and preferences of patients considering, preparing for and recovering from elective primary TKR surgery?To what extent does existing SDM research in TKR surgery incorporate Team talk, Option talk and Decision talk, as used in the model of Elwyn et al., (2017) of SDM? To what extent are the needs and preferences of patients, as found by answering RQ1, acknowledged in existing SDM research on TKR surgery? 

## 2. Materials and Methods 

Our scoping review used the five methodological steps described by Arksey et al [9]. In this review, we focus on three key moments in the care process of patients eligible for TKR, suitable for SDM: (1) the decision to undergo surgery or not, (2) the decision regarding how to prepare for surgery and (3) the decision regarding how and where to recover after surgery.

### 2.1. Search Strategy, Identification of Relevant Studies

To retrieve relevant studies, we used a broad systematic search strategy consisting of a search string that identified studies related to TKR. Subsequently a separate search string was built related to SDM. The SDM search string was based on a Cochrane review of Légaré et al. [10]. Finally we combined the search terms related to TKR and the search terms related to SDM (Appendix A; Table A1). We included published, unpublished and in-progress studies until 3 April 2020 in the electronic databases MEDLINE, Embase, CINAHL, PsychINFO and the Cochrane Database of Systematic Reviews (CDSR). Additionally, all included full-text articles were checked for useful new references. We included qualitative, cohort and experimental studies that aimed to study SDM processes in adults who were considering, preparing for or recovering from elective primary TKR written in the English language. Studies that investigated SDM in general orthopedics that did not separately treat or analyze patients making decisions regarding TKR were excluded. Literature shows that recovery trajectories and influencing factors of persons undergoing TKR are significantly different compared to those of other orthopedic procedures [11,12]. Records were managed using Endnote X8.

### 2.2. Study Selection

First, two reviewers (G.S. and J.J.) independently screened the articles by title and abstract. If the title and abstract suggested that an article was potentially eligible for inclusion, a complete hard copy of the report was obtained. Next, the same reviewers independently assessed the full text articles to determine their eligibility. We only included articles that specifically studied SDM in patients considering or undergoing TKR surgery to answer our research questions. However, a substantial part of the available literature considers total hip replacement (THR) and TKR to be similar surgical interventions. We strongly disagree with this notion, as several aspects of the surgery, underlying condition and recovery differ substantially between these procedures [13,14]. Nevertheless, some of these studies could potentially contain valuable (indirect) insights regarding SDM in patients undergoing TKR surgery [15,16,17,18,19,20,21,22,23,24]. By completely excluding these articles, we could have missed relevant information.

We did not utilize the articles that studied decision making in this mixed population of TKR and THR patients to answer our research questions. However, we purposely did select these articles to gain a complete overview (broad scope) of the main outcomes and relevant details of SDM in the field of TKR surgery. We have tabulated these studies in Appendix A
Table A7a,b and discussed differences between the included studies and these “broad scope”-studies in the Discussion Section. 

The articles that did met our eligibility criteria were included for an extended review. The flow of our search strategy is displayed in a Prisma flowchart (Figure 1). The Prisma checklist is included in the Appendix A, Table A2.

### 2.3. Methodological Assessment

GS and JJ independently assessed the rigor of the qualitative studies using the Critical Appraisal Skills Programme (CASP Checklists Oxford (2014) (Appendix A; Table A3). Rigor implies that reliability and validity should be applied to qualitative research during the inquiry rather than only to the post hoc analysis of the data [25]. The methodological quality of the quantitative studies was assessed using the Hoy’s risk of bias tool [26] (Appendix A; Table A4). Disagreements in this process between the two reviewers (G.S. & J.J.) were resolved in a consensus meeting.

### 2.4. Data Extraction

Both reviewers (GS and JJ) independently charted the data from eligible studies using a standardized data charting form. The following study characteristics were extracted from all studies: authors, type of publication and country of origin; aims/objectives of the study, study design (including control groups, if any), in- and exclusion criteria; population type and setting, eligibility criteria, number of participants and age, gender and ethnicity; elements of SDM and underlying rationale; and main findings and outcome variables of the study.

### 2.5. Data Analysis

To answer RQ1, we used the principles of meta-ethnography to synthesize data from the qualitative and quantitative studies [27]. First, GS developed the framework of concepts and themes, based on study data and pertinent discussion points. Subsequently, JJ independently reviewed the studies and further developed the framework. We used NVivo version 11 software (QSR International, Victoria, Australia) to synthesize the research themes. Finally, we checked, discussed and adjusted the derived concepts and themes for clinical meaningfulness and face validity in an iterative process of several rounds until we reached consensus with TH, MM and MNvdS, who are all experienced in qualitative research and in the scientific and practical application of SDM processes.

Second, to answer RQ2 we categorized the TKR decision studies according to the three elements of the model of Elwyn et al. (i.e., team, option and decision talk) [2]. Subsequently, we narratively described the findings of this data categorization. 

Finally, to determine to what extent the needs and preferences of patients, as found by answering RQ1, are acknowledged in existing SDM research on TKR surgery, GS and JJ independently assessed each study for how well it covered the main themes that were derived from the meta-synthesis (“fully covered,” “partly covered” or “not covered”). Any discrepancies between the assessments of the two reviewers (G.S. & J.J.) were resolved in a consensus meeting. 

## 3. Results

Our initial search yielded 3460 titles. After removing the duplicates (*n* = 934), we screened the titles and abstracts of 2526 articles. All disagreements were resolved by a consensus meeting between the two reviewers. After reading 92 potentially relevant articles, 28 articles were included for a “broad scope.” Out of these 28 articles, 17 articles were included in this review for answering the RQ’s. The other 11 articles were not included in the “extended scope” of this review because they studied a mix of patients undergoing THR and TKR surgery (Appendix A; Table A7a,b) [15,16,17,18,19,20,21,22,23,24,28].

From the 17 included articles, we found ten articles eligible for answering RQ1 (see Table 1) and seven for answering RQ2 (see Table 2). The seven studies related to RQ2 were all of quantitative nature. Four of these seven studies reported on the change in the number of performed TKR procedures as a result of using a decision aid [29,30,31,32]. Two studies reported a significant reduction in the number of surgical TKR procedures (reduction rates ranged between 14–38%) [29,30]. Stacey et al found no statistically significant reduction in the number of procedures [31]. One study assessed whether the use of a decision aid improved access to total knee replacement surgery for self-identified black patients with OA of the knee [32]. The authors found an 85% increase in surgery rates due to the use of their decision aid. Three studies researched the effect of using a decision aid on decisional conflict [30,33,34]. All three demonstrated a reduction in decisional conflict. Since the third research question connects the results of the first two research questions, no additional articles were needed to answer this research question. 

### Methodological Assessment

Of the qualitative studies (RQ1), four studies had low rigor [39,42,43,44]. and six studies had high rigor [35,36,37,38,40,41]. The interaction between researchers and patients was not mentioned in any of the high rigor studies. It was also unclear in all the high rigor studies whether ethical approval was obtained. Further details of the scores are shown in Appendix A
Table A3.

From the seven quantitative studies, four are randomized controlled trials [30,31,32,33] and three are cohort studies [29,34,45]. Four studies had a low risk of bias [30,31,32,33] and three studies had a moderate risk of bias [29,34,45]. None of the studies made it clear whether the studied population was representative of the (inter)national TKR population. Further details of the scores are shown in Appendix A
Table A4.


*RQ1. What is known regarding the decisional needs and preferences of TKR patients?*


The decisional needs and preferences of patients considering TKR were categorized into four different themes. A brief summary is presented in the following Sections and Table 3. Additional details can be found in Table A5.

Theme 1: Personal factors with the potential to impact decisions regarding TKR care

The first theme consists of three categories:

(1) Fears and concerns regarding the surgical treatment. 

Patients mentioned fear of TKR surgery, fear of anesthesia, concerns regarding postoperative pain or complications and concerns regarding long-term outcomes of TKR [35,39,40]. Fear of the operation was found to be an important reason for postponing surgery, even for patients who received the clinical advice to undertake the operation [35,44]. 

(2) Concerns and preferences of patients for candidacy or to postpone or refuse surgery. 

Older patients were more likely to postpone or refuse TKR. Factors for this preference were: patients felt too old, patients felt that they suffered from unresolved severe comorbidity and they preferred other treatment modalities such as medication or physical therapy [39]. Patients who felt that they were ready to undergo surgery often could no longer cope with the symptoms of their OA [43]. Patients also perceived that non-surgical treatments were “band-aid solutions” that could not repair the damage to the knee [43].

(3) Ethnic variability. 

In a study group of women of Arab origin, preferences regarding TKR were influenced both by the ambivalence caused by fear and lack of information regarding the potential harms and benefits of TKR as well as by the clinician’s advice about the best treatment option (and second opinions from abroad) [35]. One author found that Caucasian patients reported more willingness to undergo TKR surgery than African American and Hispanic patients [40]. 

Theme 2: External factors with the potential to impact decisions regarding TKR care

The second theme consists of two categories:

(1). Interaction between the patient and the orthopedic surgeon. 

An important factor was the patient-doctor relationship, which was universally seen by patients as a major factor in decision making [35,37,39,40]. Important patient issues in the discussion about the patient-doctor relationship were communication, information and trust [35,40]. 

(2). Issues that could enhance, delay or hinder decision making. 

The timing of decision making was affected by several factors, such as ambivalence of patients, concepts of readiness for surgery and surgery perceived as a last resort by patients [35,40]. Financial issues were often discussed by patients in the decision-making process; however, these issues would not affect patients’ final decision to undergo TKR [40].

Theme 3: Patient reliance on a variety of information sources for TKR decisions

The third theme consists of two categories:

(1). Personal experiences. 

Both positive and negative personal experiences of peers with knee osteoarthritis (OA) played a major role in patients’ attitudes and beliefs about TKR surgery and therefore both had a substantial influence on the decision-making process of patients who were considering TKR [40,44]. 

(2). Experiences of relevant others. 

Patients used different sources to obtain information, such as second opinions and general practitioners [35,38,40,44], but also non-professional contacts such as relatives and media [22]. Experiences of relatives or friends with TKR surgery played a major role in patients’ thoughts about outcome and decision making [35,36,39,40]. Patients saw their social network as an important source of information about a major surgery procedure like TKR [36].

Theme 4: Prediction tools and presentation of relevant information to enhance care decisions: 

The fourth theme consists of three categories:

(1). Value of prediction outcome tool. 

Patients valued a theoretical outcome prediction tool such as a decision aid over other potential information sources and felt that such a tool could enhance decision making [36]. 

(2). Methods to obtain relevant information. 

Decision aids that explain various orthopedic treatment choices with their risks and benefits would be helpful for patients in a decision-making process [37]. Patients preferred a bottom-line outcome prediction, presented by a prediction tool [36]. 

(3). Presentation of relevant information. 

Regarding the presentation of outcome probabilities, patients mentioned that they needed a bottom-line prediction, with visual presentations [36,38]. Patients mentioned that the presentation of the risk and benefits of the surgical procedure needed to be personalized, based on their individual characteristics [38]. 


*RQ2: To what extent does existing SDM research in TKR surgery incorporate Team talk, Option talk and Decision talk, as used in the model of Elwyn et al. (2017) of SDM?*


The elements of SDM obtained from the seven quantitative studies were categorized into Team talk, Option talk and Decision talk. In six out of seven studies, decision aids were used [29,30,31,32,33,34]. The seventh study focused on the importance of control preferences of patients who chose to undertake TKR [45]. A brief summary is presented in the following Sections.

Team talk

Team talk emphasizes supporting patients as they are made aware of choices, while also eliciting their goals as a means of guiding the decision-making process. This element was partially recognized in five articles [30,31,32,33,34]. In these five articles, decision aids were used with the aim of providing patients with insight into different options of treating knee OA [30,31,32,33,34]. Boland and Stacey also mention presurgical assessments between a healthcare professional and the patient [30,31]. However, the content of these assessments was not specified. De Achaval et al. mention that one of the researched decision aids, the Adaptive Conjoint Analysis (ACA), ranked eight characteristics in importance to the patient [33]. The characteristics were not specified in the article. The results of the ACA were displayed as bar graphs. Longer bars represented higher importance to the patient. The printed results were given to the patients and explained by a research assistant. It remains unclear how or if the decision aids mentioned in the different studies contributed to the conversation between the patient and the health care professional.

Option talk

Option talk refers to the task of comparing alternatives, using risk communication principles. Two of seven studies mentioned preoperative assessments between patients and a healthcare professional [30,31]. Four articles mentioned a discussion between the surgeon and the patient [30,31,32,33]. De Achaval et al, and Boland et al state that in the patient-physician conversation the goal is “to decide on the course of action” [30,33]. Stacey et al mentioned that the surgeon is provided with an overview of the results of the decision aid [31]. The way the results influenced the decision making is not explained. Ibrahim et al mentioned that baseline knowledge is important in patient-physician interaction [32]. None of the studies specify the structure or content of the interactions between patients and professionals. Therefore, it remains unclear if the element of Option talk was covered in these interactions.

Decision talk

Decision talk refers to the task of arriving at decisions that reflect the informed preferences of patients, guided by the experience and expertise of health professionals. The study of Stacey et al. [31], explicitly described this step. They stated that “patients need to discuss their values and preferences with the orthopedic surgeon, prior to feeling certain about the best treatment choice for them.” Arterburn et al., and Ibrahim et al., mentioned that it remains unclear whether or how the use of decision aids influenced the discussion about having surgery or not [29,32]. Boland et al, and De Achaval et al, mentioned that there is a conversation between the patient and the orthopedic surgeon, in which the decision whether or not to have surgery is made [30,33]. 


*RQ3. To what extent are patients’ needs and preferences taken into account in SDM?*


We found little evidence that patients’ needs and preferences (identified in RQ1) were addressed in the studies included for answering RQ2 (see Appendix A
Table A6). The “personal factors” and “external factors” relevant to patients’ decisions were partially addressed in three studies [30,34,35]. One study investigated the personal factor “control preference” but this was not integrated into a decision intervention [45]. Six out of seven studies considered a number of factors related to the theme “sources of information” to enhance the decision process [29,30,31,32,33,34], as they studied the impact of using specific decision aids. One study examined patient preferences regarding multiple sources of information (provider opinion, patient testimonial, outcomes prediction) and the presentation of this information [33]. 

## 4. Discussion

Our aim was to probe the available literature in order to synthesize what is known regarding both patient thoughts and preferences surrounding TKR as well as the state of the art of SDM in TKR care. We identified four themes that may be important for patients to consider for optimizing decisions related to TKR: (1) patients’ personal factors related to decision making, (2) external factors related to decision making, (3) sources of information to enhance decision making and (4) outcome prediction and presentation of relevant information (RQ1). We found that the research on SDM in TKR mainly focused on the decision to undergo surgery, not on the preparation for surgery or the postoperative care phase. In the studies that assessed the impact of a decision intervention, we found that “Team talk” was typically (partially) utilized. However, “Option talk” was not identified and “Decision talk” was mostly overlooked (RQ2). Finally, we found a discrepancy between existing decision interventions and patients’ needs and preferences, potentially resulting in suboptimal SDM (RQ3).

Regarding the decisional needs and preferences of patients, our findings are in line with two previous systematic reviews. Barlow et al., [46] reviewed qualitative studies regarding decision making in TKR surgery. Their main objective was to identify factors that influence patients’ decisions when considering TKR, which is in line with our first research question. They identified several themes relevant to the patients’ decision whether to undertake TKR surgery. The following themes overlap with the themes in our study: expectations of surgery, fear, patient-doctor relationship, social network, previous experiences with surgery, pain and functioning. They also found that psychological implications, conflict in opinions and coping mechanisms were important issues for patients in making this decision [46]. O’Neill et al, [47] concluded that for TKR the unmet needs and influencing factors for decision making are complex. Patients must consider many factors before deciding to undergo TKR. Patients point out the importance of the patient-healthcare professional relationship in this process [47]. This is also in line with our findings and emphasizes the importance of inquiring about the needs, preferences and capacity of patients before participating in decisional processes. Finally, the process by which healthcare professionals deliver information (ideally treatment options as well as harms and benefits) was not reproducibly described in most of the reviewed studies [1,48,49]. 

We revealed that in the available literature, authors have mainly studied the usefulness and effects of decision aids. Although it is evident that well-designed decision aids have the potential to increase patients’ knowledge, decrease decisional conflict and improve patient involvement in the decision-making process [1,50]. However, we were surprised to find that none of these interventions addressed all the patients’ individual needs and preferences related to personal and external factors, as identified in the answer to RQ1. Our findings suggest that a decision aid should address the patients’ individual needs and preferences related to personal and external factors, while providing a complete and up-to-date insight into the different available treatment options for the perceived problem of the patient with knee OA. Finally, such a decision aid would ideally provide understandable visual presentation of the different treatment options and the use of outcome prediction scenario, ideally adjusted to each of the treatment options per individual type of patient. 

The available literature suggests that only parts of the SDM process are studied. After all, SDM should happen within the encounter between patients and healthcare professionals [51]. The studied decision aids were often provided to patients before the encounter with the orthopedic surgeon. It remained unclear how they impacted the interaction between patient and professional within the clinical encounter. As Hargraves et al., [51] stated: “*A decision aid, patient power, medical skills and scientific evidence do not simply result in good decisions by being in a room together. Each may potentially contribute but their potential is drawn out and realized in conversation*.” Unfortunately, this last step remains largely unexamined in the available studies. However, we find it promising that two recently published study protocols have the intention to research the effect of SDM interventions that take patient characteristics and preferences into account [52,53]. After all, decisions regarding optimal patient care should jointly be made on an individual level, rather than on population level. Such shared decisions are therefore unique, based on preferences, specific environmental aspects and therefore needs careful deliberation between patient and healthcare professional [54]. The three-talk model of Elwyn et al, is an example of an established framework that helps to involve the patient in such a deliberation and the subsequent health and care decision to be made [2]. This framework contains the relevant steps and principles of SDM and presents easy to remember and execute conversational steps to optimize the conversation between patients and professional. Finally, future studies need to structurally measure the quality of the decision process of the conversations held. A good example of measuring decision quality is described in the study protocol of Mangla et al, who are examining the impact of patient-directed and physician-directed decision support strategies for patients with hip and knee osteoarthritis [55]. 

According to the principles of SDM, patients and healthcare professionals should collaborate on decisions, considering all the available treatment options (including the option to do nothing). Therefore, SDM should perhaps be an interdisciplinary responsibility in which patients and different healthcare professionals (including the orthopedic surgeon) collaborate. Together they can discuss and address all options related to the patients’ needs and preferences towards health and (physical) functioning. An interesting finding in this context is the impact that a patient’s ethnicity can have on the decision making process. There seems to be differences in preferences, needs and thoughts across different ethnical groups regarding health and disability. We believe this is an important point of attention for future developments and studies regarding SDM and their supporting frameworks or models. An important limitation of our work is that our systematic search yielded only a small number of studies, which varied widely in design, patient inclusion criteria and primary aims. This small number is partly explained by the fact that we only included articles that studied SDM in people having TKR surgery. However, a substantial part of the available literature includes both THR and TKR related research, as they are considered similar surgical interventions. We strongly disagree with this notion, as there are several aspects of the surgery, underlying condition and recovery that differ substantially between these procedures [13,14]. Nevertheless, some of the studies that combine TKR and THR groups could potentially contain valuable (indirect) insights regarding SDM in people undergoing TKR surgery [15,16,17,18,19,20,21,22,23,24].By excluding these articles, we could have missed new information. Therefore, we have extracted and presented the relevant information from these studies, alongside the results of this review, in Appendix A
Table A7a,b. The analysis of these additional articles did provide us with two additional insights. First, one study described patients’ decisional needs and preferences regarding the rehabilitation procedure after surgery instead of the decision to undergo surgery or not [15]. They describe that earlier experiences of patients, as well as the experiences of relevant others and the dominant rehabilitation regimes determine the decisional needs and preferences regarding rehabilitation [15]. And second, the study of Conner-Spady described how patients decide if they are ready to undergo surgery [18]. Assumptions about prosthesis survival, length of waiting list and the feeling that having total joint replacement would stigmatize them as being ‘old’ were found to play an important role in determining the readiness of patients to undergo surgery [18]. 

This study highlights several opportunities for future SDM research in TKR surgery. First, the focus of SDM for persons with chronic knee conditions should be broader than just the decision to undertake TKR surgery or not. Also, the precise clinical setting for the SDM process should be carefully considered. However once the decision to undergo surgery has been made, the focus of decision making should be broader as well. For instance, the decision regarding how to prepare for surgery and how to recover after surgery, that is, at home or elsewhere and to how optimize rehabilitation, should also be part of the SDM process. Moreover, the SDM interventions should take into account the patient’s personal and external factors (like, fear, coping strategies, expectations, socioeconomic status, informal network, environmental issues, etc.) regarding the healthcare decisions. Finally we recommend matching SDM methods and tools with (a) real life practice (e.g. patient and professional preferences and possibilities, contextual issues etc) and (b) theoretical concepts for optimal SDM (e.g., the model of Elwyn et al) [2]. 

## 5. Conclusions

This scoping review has uncovered specific gaps in SDM research for patients considering TKR surgery. Research on SDM for patients with chronic knee conditions seems to be in an early stage and certain steps are necessary for its advancement [56,57]. Future research should ensure the methods and tools used for SDM incorporate literature-based concepts of patients’ needs and preferences, as well as the current theoretical concepts for optimal SDM in practice. 

## Figures and Tables

**Figure 1 ijerph-18-00668-f001:**
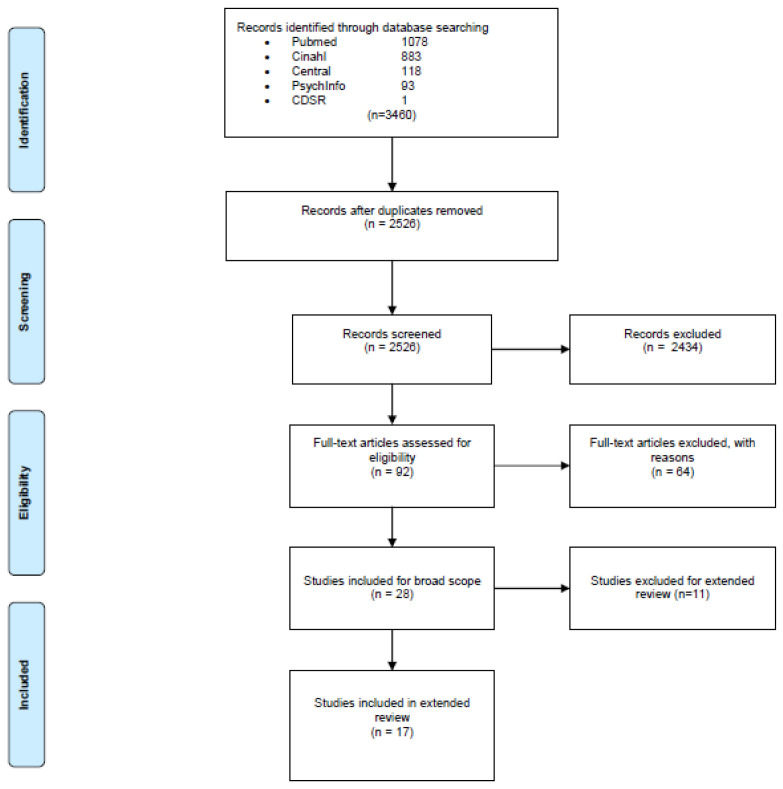
Prisma flowchart of literature search strategy.

**Table 1 ijerph-18-00668-t001:** Characteristics of the included manuscript to determine which factors support the decisional skills and / or capacity of patients considering, choosing, preparing and/or recovering from total knee replacement (TKR) surgery?

Author, Year of Publication	Population, Inclusion Criteria Participants; Age (SD, Range); Gender (%); Ethnicity	Aim of Study	Study Design	Authors Conclusions
Al Taiar, 2013 [35]	Female patients on the waiting list for TKR in Kuwait; *n* = 39; 62 (7.9); female (100%); Arabic	Study of pain experience and mobility limitations as well as the patient decision among woman on the waiting list for TKR surgery.	Qualitative study In depth interviews	Both verbal and written information about TKR should be provided as part of preoperative rehabilitation. This is critical to improve doctor-patient interactions and facilitate informed decision about the procedure and thus achieve patient-centered healthcare.
Barlow, 2016 [36]	Focus groups with patients after TKR and in depth interviews with patients considering TKR; *n* = 12 in focus group and *n* = 10 in in-depth interviews; 65,5; female (45%); British, Asian	Examination how individual predictions of outcome could affect patients decision making by providing fictions predictions to patients at different stages of treatment.	Qualitative study In depth interviews	An outcome prediction tool has the most effect targeted towards people at the start of their treatment pathway, with a “bottom line” prediction of outcome.
Ho, 2015 [37]	An elderly patient with cognitive limitations with a symptomatic right knee; *n* = 1; 77; one female; American.	Establishing the decisional capacity of elderly patients and providing a capacity adjusted approach to SDM.	Case report	With respect for autonomy demands support for patient participation and decision making in their own care, many elderly patients demonstrate questionable understanding and/or desire in making healthcare decisions.
Kesternich, 2016 [38]	Hypothetical patients diagnosed with knee OA; n/a; n/a; n/a.	To analyze the effect of personalized outcome probabilities on treatment decisions.	Qualitative study Internet survey	Patient specific outcome forecasts significantly influenced decisions with effect sizes comparable to those of physicians opinion and patient testimonials.
Yeh, 2016 [39]	Older adults (1) diagnosed with knee OA and recommended by their physicians to undergo TKR, (2) in decision about the surgery, (3) 60 years old and (4) able to communicate; *n* = 26; 73.6 years old (SD 1⁄4 6.9, range 1⁄4 61–86); female (76.9%); Taiwanese.	To explore factors related to the indecision of older adults with knee osteoarthritis (OA) about receiving physician-recommended total knee arthroplasty (TKR) and their needs during the decision-making process.	Qualitative study In depth interviews	Subjects were undecided about whether to undergo physician-recommended TKR due to treatment-related, physical condition-related, surgery-related and postsurgical care concerns. When a TKR is recommended, physicians must also educate patients about preparations for surgery, postsurgical care, rehabilitation and medicines while they are deciding whether to undergo knee-replacement surgery.
Suarez-Almazor, 2010 [40]	Physician diagnosis of knee osteoarthritis; no previous knee replacement; race (African-American and non-Hispanic, Hispanic or white and non-Hispanic); age (55 to 80 years); *n* = 37; n/a; female (62.1%); 13 White, 15 African-American, 9 Hispanic.	To conduct a qualitative analysis of decision-making factors influencing preferences for TKR in patients with knee osteoarthritis.	Qualitative study In depth interviews	Patient experiences, fears and expectations and physician trust are prominent factors influencing decision making. An open doctor-patient is important to achieve satisfactory shared decision-making for TKR. Doctor-patient interactions and subsequent patient decision-making could be improved by developing and using decision aids for patients and educating physicians about patient concerns and expectations.
Kroll, 2007 [41]	Inclusion: physician diagnosis of knee OA, no previous knee replacement, self-reported ethnic background African American non-Hispanic, white non-Hispanic, age 55–80, English language proficiency; *n* = 37; 64 (no SD); female (62.1%); African American non-Hispanic, Hispanic, white non-Hispanic.	To identify decision making factors influencing ethnic preferences for TKR in patients with knee OA.	Qualitative study	Patient attitudes and beliefs vary among ethnic groups. There is a need for open patient-doctor communication around individual experiences and beliefs in an effort to enhance decision making for TKR.
Barlow, 2018 [42]	Focus group: *n* = 12 Interviews: *n* = 10. Inclusion: n/a, exclusion n/a. Focus group: 71,75 (n/a, n/a), female (58.33%), white *n* = 11, Indian *n* = 1. Interview group: 64 (n/a, n/a), female (30%), white (*n* = 9), Asian *n* = 1.	To explore the factors that affect decision making in TKR surgery, to help understand patients’ decision-making, which is critical in informing patient-centered care. These can be used to enhance decision-making and dialogue between clinicians and patients, allowing a more informed choice.	Qualitative study In depth interviews	An awareness of the deliberation phase, the factors that influence it, the stress associated with it, preferred models of care and the influence of the decision-making threshold will aid useful communication between doctors and patients.
O’Brien 2019 [43]	Patients on a waiting list to undergo TKA (*n* = 27) Female 48.1%, age over 70: 44.4% BMI > 30 kg/m2: 59.3% TKR contralateral: 48.1%	To explore patient factors that impact to the decision to progress to TKR, including experiences in general practice, perceptions of their condition and the access to community based allied health services	Qualitative investigation using semi structured interviews	Analyzing patients’ experiences highlighted missed opportunities in general practice to orient patients to try first non-surgical interventions. Patients require improved support to navigate allied health services
Hsu 2018 [44]	Older adult patients (*n* = 79) scheduled for TKR within 1 month Female 74.7% Mean age 71.6 years (6.8) Previous TKR: 24.1%	To explore triggers of and decision making patterns for older adults with knee OA to receive TKR	Qualitative study Data were collected in individual interviews using a semi structured guide	Main triggers to receive TKR in older adults were severe pain and inability to walk. Four decision making patterns were identified: surgery as last choice, previously receives TKR, perceived one as young and wanted to enjoy life and adjusted work characteristics but in vain.

Abbreviations: n: number, n/a: not applicable, OA: osteoarthritis, SD: standard deviation, SDM: Shared decision making, TKR: Total knee replacement.

**Table 2 ijerph-18-00668-t002:** Characteristics of the included manuscript for RQ2: How are shared decision making (SDM) processes supported among patients, regarding the three key decision moments before and after TKR.

Authors, Year of Publication	Population, Inclusion Criteria; Number of Participants; Age (SD, Range); Gender (%); Gender; Ethnicity.	Aim of Study	Study Design	Main Findings
Arterburn, 2012 [29]	Patients with knee or hip osteoarthritis (ICD-9), over 45 years of age; *n* = 3510; 65.0 (11.1); female (62%); n/a.	To examine the associations between introducing decision aids for elective hip and knee replacement and changes in rates of surgery and costs of care.	Observational study	The introduction of decision aids was associated with 38% fewer knee replacements and 12-21 % lower costs over 6 months. Decision aids: Evidence based video. Evidence based written information. Goal of the decisions aid: not explicitly stated.
Filardo, 2017 [45]	Patients who underwent TKR between 2011 and 2015 in one hospital in Italy; *n* = 176; 66 (9); female (68.2%); n/a	To evaluate if a more active role in the patient decision making preference may be correlated with a more successful outcome in patients undergoing TKR.	Observational study.	The control preference of patients undergoing TKR is correlated with the final outcome. Decision aids: non described.
de Achaval, 2012 [33]	Patients medically appropriate for a TKR; *n* = 208; 62.8 (9.0); female (68%); 66% white, 24% black, 7% Hispanic and 3% other.	To evaluate the impact of different decision aids, on patients’ decisional conflict associated with TKR surgery.	Randomized controlled trial.	Audio-visual patient decision aid decreased decisional conflict more than printed material alone or than the addition of a more complex ACA tool. Decision aids: Printed booklet Video booklet + printed booklet Video booklet + ACA tool Goal of the decision aids: to increase knowledge about risks and benefits of therapeutic alternatives, to help clarify values and preferences, to prepare for the encounter with the physician and deciding on the course of action.
Ibrahim, 2016 [32]	People, self-identified as black person with frequent knee pain and over 50 years of age; *n* = 304; 59.1 (7,2); female (51%); Black.	To assess whether a decision aid improves access to total knee replacement surgery for black patients with OA of the knee.	Randomized controlled trial.	The use of a knee decision aid increased the receipt of TKR within 12 months by 85%, compared to the control group. Decision aid: video that provides information about different treatment options (risk, benefits, known efficiency), as well as information about surgery (indications, duration of surgery and hospital admission, need for rehabilitation and physical therapy, recovery time and effort, cost, risk of surgery). Goal of decision aid: to increase relevant knowledge.
Volkmann, 2015 [34]	Eligible participants were between 55–85 years of age, able to speak and read English and had moderate to severe knee OA, (score of >39 on the WOMAC). Exclusion criteria: included: ≥3 Charlson comorbidity index or a single specific comorbidity (dementia, stroke with residual plegia or paresis, cancer (other than skin) and/or end-stage liver disease. Patients reporting a history of inflammatory arthritis, recent significant knee trauma, residence in a nursing home or prior hip or knee replacement surgery were also excluded; *n* = 111; female 72 (8.2), male 70 (9.6); female (63.1%); n/a.	To examine the impact of exposure to a decision aid on changes in expectations of health outcomes following TKR and to evaluate decision-making parameters of the decision aid among men and women with knee OA.	Observational study.	A decision aid has the potential to improve post-TKR expectations. It may be beneficial reducing gender disparities in TKR patients. Decision aid: Video with evidence based information. Personalized arthritis report. Goal of decision aid: to provide relevant information and increase knowledge.
Stacey, 2014 [31]	Eligible knee OA patients were those with access to a television with a VCR or DVD player. Exclusion: inflammatory arthritis, previous TJR, uncorrected hearing or visual impairment or unable to read or understand English; *n* = 142; Intervention 76.1 (10.85) control 67.3 (12.16); female (67.7%); n/a.	To evaluate feasibility and to provide preliminary data on the effectiveness of a decision aid with a preference report for surgeons on wait times and decision quality in patients with OA considering TKR.	Pilot randomized controlled trial.	It was feasible to recruit patients with knee osteoarthritis, administer the decision support interventions and collect outcome measures. Preliminary effectiveness outcomes demonstrated that the used decisional aid was associated with less waiting time, lower surgery rates and improved decision quality and knowledge. Decision aid: DVD Booklet Goal of the decision aid: to inform patients about surgery and non-surgical options.
Boland, 2018 [30]	Inclusion: moderate to severe knee OA. Exclusion: inflammatory arthritis, previous total joint arthroplasty surgical consultation, unable to read or understand English or no access to a television with a VCR/DVD player to view decision aid. *n* = 242; 65 ( 10.3, n/a), 67 ( 9.2, n/a) 69 ( 8.2, n/a), 67 (7.8, n/a); female (59.99%); n/a.	To gather more knowledge, in order to better understand the circumstances that optimize the use of decision aids.	A subgroup analysis of a larger prospective 2-site randomized controlled trial.	The decision aid had a greater effect at the academic site than at the community site, which provided longer consultations with more verbal education. Hence, decision aids might be of greater value when more extensive total knee arthroplasty pre-surgical assessment and counselling are either impractical or unavailable. Decision aids: Video Booklet Clinic specific information about pre-rehabilitation. Goal of the decision aid: to provide information.

Abbreviations: ACA: Adaptive Conjoint Analysis, DVD: Digital Versatile Disc, ICD: International Statistical Classification of Diseases and Related Health Problems, n: number, n/a: not applicable, OA: osteoarthritis, SD: standard deviation, SDM: Shared decision making, TJR: total joint replacement, TKR: total knee replacement, VCR: videocassette recorder, WOMAC: Western Ontario and McMaster Universities Osteoarthritis Index.

**Table 3 ijerph-18-00668-t003:** Themes and categories relevant to patients for participating in a SDM process around total knee replacement surgery.

	Al Taiar 2013	Barlow 2016	Ho 2015	Kesternich 2016	Yeh 2016	Suarez 2010	Kroll 2007	Barlow 2018	O’Brien 2019	Hsu 2018
**Themes**/*categories*										
**Theme 1: Personal factors with the potential to impact decisions regarding TKR care**										
*Fears and concerns regarding the surgery*										
*Concerns and preferences of candidacy or refuse surgery*										
*Ethnic variability*										
**Theme 2: External factors with the potential to impact decision regarding TKR care**										
*Interaction between patient and orthopedic surgeon*										
*Issues that could enhance, delay or hinder decision making*										
**Theme 3: Patient reliance on a variety of information sources for TKR decisions**										
*Personal experiences*										
*Experiences of relevant others*										
**Theme 4: Prediction tools and presentation of relevant information to enhance care decision**										
*Value of prediction outcome tool*										
*Methods to obtain relevant information*										
*Presentation of relevant information*										

Abbreviations: SDM: shared decision making, TKR: total knee replacement. Grey Background color: Indicates the presence of a category in a study.

## Data Availability

Data sharing not applicable No new data were created or analyzed in this study. Data sharing is not applicable to this article.

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
