# Peer review of "Current Status and Future Prospects for Shared Decision Making before and after Total Knee Replacement Surgery—A Scoping Review"

_ijerph, 2021, doi:10.3390/ijerph18020668_

Round 1
Reviewer 1 Report
In this review, van der Sluis et al. survey a large body of existing literature related to patient and health care team decision to pursue elective total knee replacement (TKR) surgery. Their choice of topic identifies a unique area of medicine where the decision to pursue or not pursue surgery can be largely made by the patient once offered by a health care professional and involves significant considerations of the magnitude of surgery and potential outcomes. The need for pertinent discussions, decision aid tools and recognizing patient considerations is evident across medicine but this topic provides a great opportunity for intervention and alternate decisions to be made. The authors have done a thorough job identifying the relevant literature and synthesizing the information to answer three main research questions. Their summary of the content provides helpful insight into the current status of research surrounding shared decision making and the need for further investigation. It also provides valuable insight into the important patient considerations when making a health care decision which could help guide both future research as well as physician practices for approaching patient conversations. I believe their review provides a valuable addition to the literature and find it acceptable for publication with only the following minor changes.
Comments:
- Scoping review or systematic review? These are two distinct categories. And a ‘systematic scoping review’ is not proper terminology. Scoping review still requires a systematic approach but describes the way in which the results are presented in the text.
It seems this is a scoping review, as well as based on the references cited, please remove the word systematic. (Page 1, line 31; Page 2, line 91; )
- It is unclear to me what the total 28 articles identified to meet criteria for “broad scope” were used for. It seems the RQs were answered solely using the 17 extended review articles so it would be helpful to include in the methods what components the broad scope was considered for. Page 4 lines 160-. It seems they are only reported in Table A6 and mentioned in the discussion, if so, I think it would be worth stating this earlier on.
- The reported ethnical variability in RQ1 was an interesting finding that was not the primary objective of the study. It could be an interesting short discussion point to expand upon how best to approach this in future studies should the authors choose to do so.
- There was a significant amount of information collected regarding RQ1. Since the main objective of this question was to synthesize what is known regarding patients thoughts and preferences surrounding TKR the organization and presentation of this data is of key importance. While the results section and Table 3 provide all the relevant details it would be helpful to have an overarching Figure or Summary Figure which allows for a quick identification of the 4 themes and the key words explaining the content that will be described for each theme.
- The use of decision aids seems like an important component of RQ2 and would be a valuable consideration for any future research. It would be helpful to have a bit more description on what a “decision aid” is in the context of these studies and some specific examples of what is used or could be used.
- The authors make the interesting observation that the details of how the conversations were held and the process by which health care information was delivered was not described in any of the studies. In the discussion they briefly discuss the impact of this and its importance for guiding shared decision making. It may be prudent to discuss some ways in which this could be rectified in future studies such as the use of a systematic approach to patients conversations with a pre-established framework or model.
- Overall, the authors have accurately reviewed and reported the relevant literature. One of their conclusions is that future research should be pursued for the shared decision making model. They provide some general considerations for future research in their last discussion paragraph but it would be beneficial if they could expand with more specific suggestions after their current expertise from reviewing the field to best guide ongoing research.
Author Response
Rebuttal: Manuscript ID ijerph-1062562 , entitled “Current status and future prospects for shared decision making before and after total knee replacement surgery: a scoping review
We want to thank both reviewers for both their kind words as well as their critical feedback regarding our manuscript. We believe their input has made our manuscript more readable, concrete and interesting. Please find our point-by-point responses to the reviewers’ questions and remarks on the following pages.
Reviewer 1
Comment 1: Scoping review or systematic review? These are two distinct categories. And a ‘systematic scoping review’ is not proper terminology. Scoping review still requires a systematic approach but describes the way in which the results are presented in the text. It seems this is a scoping review, as well as based on the references cited, please remove the word systematic. (Page 1, line 31; Page 2, line 91; )
Reply to comment 1: We agree with the reviewer that a “systematic scoping review” is confusing. We therefore removed the term “systematic” in relation to the study design from the manuscript.
Comment 2: It is unclear to me what the total 28 articles identified to meet criteria for “broad scope” were used for. It seems the RQs were answered solely using the 17 extended review articles so it would be helpful to include in the methods what components the broad scope was considered for. Page 4 lines 160-. It seems they are only reported in Table A6 and mentioned in the discussion, if so, I think it would be worth stating this earlier on.
Reply to comment 2: It is correct that we used the 17 articles to answer our RQs. These 17 studies specifically included patients opting for total knee replacement. However, by only looking at these studies, we would discard all studies that included both patients opting for total knee replacement and total hip replacement. We were convinced that literature about the same topic in another (in this case a broader) patient population could contain additional relevant and valuable information and insights and/or confirm our initial findings. Therefore, we used the total 28 articles which are part of the “broad scope” to check our initial review findings, in order to gain a complete overview of the relevant literature.
As stated in the Discussion, we did find additional insights (page 14, lines 403 - 413). For reasons of readability we have presented the outcomes of our “broad scope” primarily in the appendix of the manuscript. We agree that the explanation in the methods section was too concise and therefore we added the following text to the methods section:
“We did not utilize the articles that studied decision making in this mixed population of TKR and THR patients to answer our research questions. However, we purposely did select these articles to gain a complete overview (broad scope) of the main outcomes and relevant details of SDM in the field of TKR surgery. We have tabulated these studies in appendices A7 a & b and discussed differences between the included studies and these “broad scope”-studies in the Discussion section.” -- (page 3, line 122-127)
Comment 3: The reported ethnical variability in RQ1 was an interesting finding that was not the primary objective of the study. It could be an interesting short discussion point to expand upon how best to approach this in future studies should the authors choose to do so.
Reply to comment 3: We agree that this an interesting finding and thank the reviewer for the suggestion to expand our discussion section with a short statement. We added the following text to the Discussion:
“An interesting finding in this context is the impact that a patient’s ethnicity can have on the decision making process. There seems to be differences in preferences, needs and thoughts across different ethnical groups regarding health and disability. We believe this is an important point of attention for future developments and studies regarding SDM and their supporting frameworks or models.” -- (page 14 lines 388-392)
Comment 4: There was a significant amount of information collected regarding RQ1. Since the main objective of this question was to synthesize what is known regarding patients thoughts and preferences surrounding TKR the organization and presentation of this data is of key importance. While the results section and Table 3 provide all the relevant details it would be helpful to have an overarching Figure or Summary Figure which allows for a quick identification of the 4 themes and the key words explaining the content that will be described for each theme
Reply to comment 4: We agree with the reviewer that a Summary table would be helpful for the quick identification of the 4 themes. To do so, we have moved the original Table 3 to the appendix (Table A5) and have added a more concise Table tabulating the 4 themes and their respective categories (Table 3 on page 11). Please find the new table at the back-end of this rebuttal.
Comment 5: The use of decision aids seems like an important component of RQ2 and would be a valuable consideration for any future research. It would be helpful to have a bit more description on what a “decision aid” is in the context of these studies and some specific examples of what is used or could be used
Reply to comment 5: We agree with the reviewer that a more detailed description of the decision aids used in the included studies would be a valuable addition to our manuscript. Therefore we now provide additional information on the nature and goal of the decision aids in the Main Findings column of Table 2. Furthermore, we have added the following statement to our discussion session regarding the content of a future decision aid, based on the findings stemming out of/provided by our review:
“Our findings suggest that a decision aid should address the patients’ individual needs and preferences related to personal and external factors, while providing a complete and up-to-date insight into the different available treatment options for the perceived problem of the patient with knee OA. Finally, such a decision aid would ideally provide understandable visual presentation of the different treatment options and the use of outcome prediction scenario, ideally adjusted to each of the treatment options per individual type of patient.” -- (page 14 lines 356-362)
Comment 6: The authors make the interesting observation that the details of how the conversations were held and the process by which health care information was delivered was not described in any of the studies. In the discussion they briefly discuss the impact of this and its importance for guiding shared decision making. It may be prudent to discuss some ways in which this could be rectified in future studies such as the use of a systematic approach to patients conversations with a pre-established framework or model.
Reply to comment 6: Based on the reviewer’s suggestion we have now added the following information to our discussion section regarding the use of a framework to when studying SDM:
“After all, decisions regarding optimal patient care should jointly be made on an individual level, rather than on population level. Such shared decisions are therefore unique, based on preferences, specific environmental aspects and therefore needs careful deliberation between patient and healthcare professional.55 The three-talk model of Elwyn et al is an example of an established framework that helps to involve the patient in such a deliberation and the subsequent health and care decision to be made.2 This framework contains the relevant steps and principles of SDM and presents easy to remember and execute conversational steps to optimize the conversation between patients and professional. Finally, future studies need to structurally measure the quality of the decision process of the conversations held. A good example of measuring decision quality is described in the study protocol of Mangla et al, who are examining the impact of patient-directed and physician-directed decision support strategies for patients with hip and knee osteoarthritis.54” -- (page 15 lines 371-382)
Comment 7: Overall, the authors have accurately reviewed and reported the relevant literature. One of their conclusions is that future research should be pursued for the shared decision making model. They provide some general considerations for future research in their last discussion paragraph but it would be beneficial if they could expand with more specific suggestions after their current expertise from reviewing the field to best guide ongoing research.
Reply to comment 7: We have now extended the text in our last discussion paragraph and added suggestions for future research in the additional text regarding comment 5 and 6.
“Finally we recommend matching SDM methods and tools with a) real life practice (e.g. patient and professional preferences and possibilities, contextual issues etc. ) and b) theoretical concepts for optimal SDM (e.g., the model of Elwyn et al)2.” -- (page 16 lines 421-423)
Table 3: Themes and categories relevant to patients for participating in a SDM process around total knee replacement surgery.
|
Al Taiar 2013 |
Barlow 2016 |
Ho 2015 |
Kesternich 2016 |
Yeh 2016 |
Suarez 2010 |
Kroll 2007 |
Barlow 2018 |
O'brien 2019 |
Hsu 2018 |
|
|
Themes / categories |
||||||||||
|
Theme 1: Personal factors with the potential to impact decisions regarding TKR care |
||||||||||
|
Fears and concerns regarding the surgery |
||||||||||
|
Concerns and preferences of candidacy or refuse surgery |
||||||||||
|
Ethnic variability |
||||||||||
|
Theme 2: External factors with the potential to impact decision regarding TKR care |
||||||||||
|
Interaction between patient and orthopaedic surgeon |
||||||||||
|
Issues that could enhance, delay or hinder decision making |
||||||||||
|
Theme 3: Patient reliance on a variety of information sources for TKR decisions |
||||||||||
|
Personal experiences |
||||||||||
|
Experiences of relevant others |
||||||||||
|
Theme 4: Prediction tools and presentation of relevant information to enhance care decision |
||||||||||
|
Value of prediction outcome tool |
||||||||||
|
Methods to obtain relevant information |
||||||||||
|
Presentation of relevant information |
Abbreviations: SDM: shared decision making, TKR: total knee replacement
Reviewer 2 Report
This paper does a great job of being a Scoping paper to understand what the literature says about shared decision models in orthopedic total knee replacement literature.
The authors have correctly used the framework of “scoping review” rather than “systematic review” given that their goal is to better understand the state of research and literature for shared decision-making in total knee replacement. They set their three research questions and search criteria a priori. They searched multiple databases to cast a broad net and then appropriately narrowed articles down using previously agreed-upon eligibility criteria. Two reviewers independently screened abstracts, independently assessed rigor and quality of included articles, independently charted data from eligible studies and then use the whole authorship team to gain consensus around themes identified.
I have reviewed their search strategies used in the different journal databases and found them to be appropriate.
They used appropriate and acceptable criteria for study rigor (CASP) and bias (Hoy). They used a standard charting checklist for each article that was included for developing themes and lessons learned for shared decision-making in total knee replacement.
The references appear complete and appropriate.
I have no major concerns and the author team deserves much praise as I am sure this paper took a lot of work and energy and provides much needed information in this field.
I have one clarification request and one minor edit:
Table 3 has some opportunities for clarity/consistency between the table and the body of the article ( or maybe I am misunderstanding):
Table 3 references the body of the article that describes 4 themes and associated categories. However, table 3 seems to call categories “nodes”. I’m not clear of the difference and why not call them the same? Also the names of some of the “nodes” or categories are slightly different comparing table 3 and article body. For example, line 208 lists theme 1 “Personal factors” – “Candidacy”, yet table 3 says “factors related to the communication with a healthcare professional related to concerns and other preferences” rather than “candidacy”. Line 221 reads theme 2 category 1 “Interaction between the patient and orthopedic surgeon”, yet table 3 says “Factors regarding healthcare professional interaction”. Table 3 lists “Decision tools” under theme “Sources of information to enhance shared decision making” and in article body it is not there but presumably in the theme 4 “Prediction tools”. Table 3 theme 4 “Prediction tools” has node names different than the category anmes in the body of the article. Again, it appears that table 3’s purpose is to provide clear detailed examples from the literature of these themes and the body of the article (lines 197-257) is to provide summaries but the problem is the names of the categories/nodes are not consistent. It would be better for the reader to have consistency between table 3 and the body of the article for these themes and categories.
Minor edit: line 367 “an” should be “and” so it reads “TKR and THR”
Line 372 &376 “firstly” seems rarely to be used although it is technically ok. Consider going with “first” and “second”
Author Response
Rebuttal: Manuscript ID ijerph-1062562 , entitled “Current status and future prospects for shared decision making before and after total knee replacement surgery: a scoping review
We want to thank both reviewers for both their kind words as well as their critical feedback regarding our manuscript. We believe their input has made our manuscript more readable, concrete and interesting. Please find our point-by-point responses to the reviewers’ questions and remarks on the following pages.
Comment 1: Table 3 has some opportunities for clarity/consistency between the table and the body of the article ( or maybe I am misunderstanding):
Table 3 references the body of the article that describes 4 themes and associated categories. However, table 3 seems to call categories “nodes”. I’m not clear of the difference and why not call them the same? Also the names of some of the “nodes” or categories are slightly different comparing table 3 and article body. For example, line 208 lists theme 1 “Personal factors” – “Candidacy”, yet table 3 says “factors related to the communication with a healthcare professional related to concerns and other preferences” rather than “candidacy”. Line 221 reads theme 2 category 1 “Interaction between the patient and orthopedic surgeon”, yet table 3 says “Factors regarding healthcare professional interaction”. Table 3 lists “Decision tools” under theme “Sources of information to enhance shared decision making” and in article body it is not there but presumably in the theme 4 “Prediction tools”. Table 3 theme 4 “Prediction tools” has node names different than the category names in the body of the article. Again, it appears that table 3’s purpose is to provide clear detailed examples from the literature of these themes and the body of the article (lines 197-257) is to provide summaries but the problem is the names of the categories/nodes are not consistent. It would be better for the reader to have consistency between table 3 and the body of the article for these themes and categories.
Reply to comment 1: We thank the reviewer for the valuable suggestions and feedback regarding Table 3. For the understanding of the significant amount of information regarding RQ1 we decided -- also based on a suggestion of another reviewer -- to move this table to the appendix (now Table A5) and replace it for a more clear and concise summary table (Table 3 on page 11). Nevertheless, we have adjusted the relevant texts in Table A5 according to the reviewer’s suggestions (see Table A5 on pages 22-24). We have added Table 3 at the back-end of this rebuttal.
Comment 2: Minor edit: line 367 “an” should be “and” so it reads “TKR and THR”
Line 372 &376 “firstly” seems rarely to be used although it is technically ok. Consider going with “first” and “second”
Reply to comment 2: We thank the reviewer for these suggestions and we have made these minor edits in our manuscript.
Table 3: Themes and categories relevant to patients for participating in a SDM process around total knee replacement surgery.
|
Al Taiar 2013 |
Barlow 2016 |
Ho 2015 |
Kesternich 2016 |
Yeh 2016 |
Suarez 2010 |
Kroll 2007 |
Barlow 2018 |
O'brien 2019 |
Hsu 2018 |
|
|
Themes / categories |
||||||||||
|
Theme 1: Personal factors with the potential to impact decisions regarding TKR care |
||||||||||
|
Fears and concerns regarding the surgery |
||||||||||
|
Concerns and preferences of candidacy or refuse surgery |
||||||||||
|
Ethnic variability |
||||||||||
|
Theme 2: External factors with the potential to impact decision regarding TKR care |
||||||||||
|
Interaction between patient and orthopaedic surgeon |
||||||||||
|
Issues that could enhance, delay or hinder decision making |
||||||||||
|
Theme 3: Patient reliance on a variety of information sources for TKR decisions |
||||||||||
|
Personal experiences |
||||||||||
|
Experiences of relevant others |
||||||||||
|
Theme 4: Prediction tools and presentation of relevant information to enhance care decision |
||||||||||
|
Value of prediction outcome tool |
||||||||||
|
Methods to obtain relevant information |
||||||||||
|
Presentation of relevant information |
Abbreviations: SDM: shared decision making, TKR: total knee replacement